# Chlorination Is Ineffective at Eliminating Insects from Wastewater: A Case Study Using *Ceratitis capitata*

**DOI:** 10.3390/insects16121213

**Published:** 2025-11-28

**Authors:** Flora Kafunda, Arnaud Blanchet, Gaylord A. Desurmont

**Affiliations:** 1European Biological Control Laboratory (EBCL USDA-ARS), 34980 Montferrier-sur-lez, France; florakafunda@gmail.com (F.K.);; 2Department of Biological Systems and Agricultural Engineering, Iowa State University, Ames, IA 50011, USA

**Keywords:** biosafety, biocidal resistance, environmental safety, chlorine disinfection, arthropod management

## Abstract

Getting rid of insects from small bodies of water without causing harm to the environment is a global challenge. Chlorine is still used by some insect containment facilities to treat their wastewater, even though chlorine byproducts can be harmful to human health, and its real effectiveness against insects has been understudied. In this study, the efficacy of chlorine at killing insect eggs in water was evaluated by soaking eggs of the Mediterranean fruit fly, a common pest, by soaking them in water with different chlorine levels for up to two days. The longer the eggs stayed in chlorine and the higher the chlorine concentration, the fewer eggs survived. However, even at the strongest chlorine level, complete mortality was never achieved, and low chlorine levels had almost no effect. This study demonstrates that chlorination is not a reliable way to eradicate insect eggs in wastewater. Safer and more effective methods, such as heat treatment, are preferable.

## 1. Introduction

Chlorination is the process of adding chlorine or chlorine compounds to water, typically to disinfect it by killing bacteria, viruses, and other harmful microorganisms. Due to its high efficiency, inexpensive nature, and ease of application, chlorination is the most important process for ensuring the microbiologic safety of water supplies worldwide [1]. However, heavy reliance on this method has raised legitimate environmental and human health concerns, mainly due to the residual toxicity of chlorine and chlorination-associated byproducts such as trihalomethanes, haloacetic acids, and other halogenated compounds. These byproducts have been linked to mutagenic and carcinogenic effects in humans and wildlife. In addition, chlorine’s high reactivity can contribute to oxidative stress in exposed organisms and degrade water quality. For these reasons, the use of chlorination should not be trivialized [2,3,4].

Although the effectiveness of chlorination in eliminating microorganisms from water is undisputed, its insecticidal properties have been less explored. Application of chlorine-based solutions has been deemed an effective method to kill eggs and larvae of mosquitoes [5,6,7] as well as chironomid larvae [8,9]. A broader view of the effectiveness of chlorine in eliminating other groups of arthropods at different life stages is currently lacking. This knowledge would be particularly valuable for arthropod containment buildings such as quarantine research facilities or mass-rearing insectaries, which must ensure that their wastewater is free of live insects to prevent risks of escape. Exotic insects escaping quarantine facilities could establish invasive populations and pose human health concerns as well as ecological or agricultural threats [10,11]. Arthropod eggs are particularly crucial to investigate because they are more difficult to detect than other life stages in wastewater and may remain viable for days/weeks when immersed in water [12,13], two factors that could facilitate their accidental escape from containment buildings. To this day, chlorination is still sometimes used by arthropod quarantine facilities as a means to treat their liquid effluents and guarantee they are insect-free before discharge (P. Reynaud, personal communication, 2025).

Here, we evaluated the efficacy of chlorine as an insect ovicide by immersing insect eggs in solutions containing different concentrations of chlorine, ranging from 0 to 65,000 ppm, for durations of 4, 24, and 48 h, and measured egg survivorship. We used the Mediterranean fruit fly *Ceratitis capitata* (Diptera: Tephritidae), a major pest of various fruits worldwide [14], as a test species. This species has been the target of several sterile insect technique (SIT) programs for eradication or control [15], and has thus a history of mass rearing in containment facilities, making it an ideal test species for this study [16]. Many species of Tephritid flies are pests of commercial fruits, and additional knowledge on the methods that may help their elimination could have implications for sanitation purposes. Our primary objective was to determine the concentration of chlorine and the immersion duration needed to ensure the full eradication of viable eggs from a water solution. Preliminary tests as well as existing literature [17] showed that chlorine concentrations recommended for standard chlorination and shock treatments (50–300 ppm) had little effect on *C. capitata* eggs. For this reason, higher concentrations (1600–65,000 ppm) were used in the study.

## 2. Materials and Methods

### 2.1. Insect Rearing and Egg Collection

The insects used for the study originated from a rearing of *C. capitata* maintained routinely at the European Biological Control Laboratory (EBCL USDA ARS) following a protocol described in detail elsewhere [18]. Briefly, adults of *C. capitata* were kept in large cohorts in rectangular cages (15 cm × 59 cm × 19 cm, L × l × h) in a weather-controlled greenhouse (23 ± 1.5 °C; 45% R.H.) with a 12:12 L:D photoperiod. Two opposite sides of the cages were covered with fine-mesh nylon to provide an oviposition surface for females. Females deposited eggs directly through the nylon screen, and eggs fell into a container containing mineral water (Société des Eaux de Volvic, Volvic, France). Eggs of *C. capitata* do not float and typically sink in the water. Immersion in water halts the development of eggs and can be used in mass rearing facilities as a method for egg storage and synchronization of egg hatch [19]. Eggs were transferred to a 50 mL centrifuge bottle containing mineral water and stored until needed for experimental purposes. All eggs used for the experiment were 24–48 h old.

### 2.2. Chlorine Solution Preparation and Experimental Design

Chlorine solutions were prepared using two commercially available products, a liquid chlorine solution with 9.6% sodium hypochlorite (CHLOR-O 9.6% Acti Expert, SCP France, Le Monastère, France) and a calcium hypochlorite powder with 65% active chlorine (HTH shock, Arch Water Products France, Amboise, France). For each product, dilutions were made to prepare separate solutions of increasing chlorine concentrations. With the liquid chlorine product, the following concentrations were used, rounded to the nearest digit: 0 parts per million (ppm), 1600 ppm, 7490 ppm, 13,900 ppm, 24,280 ppm, 38,760 ppm, and 55,200 ppm. With the chlorine powder product, the following concentrations were used: 1625 ppm, 16,250 ppm, and 65,000 ppm. Eggs were immersed in each chlorine solution for 4, 24, and 48 h. Five replicates were used for each chlorine product/chlorine concentration/exposure duration combination, totaling 150 tests conducted (5 replicates × 3 exposure durations × 10 concentrations).

Tests were conducted using 10 mL well plates (24-well plate 10 mL, Group Dutscher Europe, Brussels, Belgium). A volume of 1 mL of eggs suspended in mineral water was pipetted into each well, followed by the addition of 4 mL of the appropriate chlorine solution. After the designated exposure time, eggs were rinsed three times using fresh mineral water in a funnel sieve (mesh size: 1 µm) to ensure complete removal of residual chlorine. The rinsed eggs were transferred to 15 mm diameter Petri dishes lined with Whatman black filter paper moistened with mineral water to prevent desiccation. Petri dishes were sealed with Parafilm (Parafilm M, Amcor PLC, Zürich, Switzerland) and incubated for 5 days, a time period sufficient for complete egg hatch [19], in a rearing room maintained at 22 °C, 65–70% RH and a 12:12 L:D photoperiod. Petri dishes were examined after 5 days under a stereomicroscope (Meiji 1.5, Meiji Techno America^®^, San Jose, CA, USA) and the numbers of larvae and unhatched eggs were counted. The sum of unhatched eggs and live larvae was used as an estimate of the number of eggs initially present at the start of the test, and % egg survivorship was calculated as number of larvae/(number of larvae + number of unhatched eggs) × 100. The average number of *C. capitata* eggs per test was 88.3 ± 3.3 (*n* = 150), and a total of 13,249 eggs were used across all treatments and replicates.

### 2.3. Statistical Analysis

The effects of chlorine concentration and immersion duration on egg survivorship were analyzed using an ANCOVA model including chlorine concentration as a continuous independent variable and duration of immersion as a categorical independent variable (three levels: 4 h, 24 h, 48 h), as well as the interaction between the two terms (α = 0.05) (JMP19, SAS Institute Inc., Cary, NC, USA). The chlorine product used (liquid chlorine solution or calcium hypochlorite powder) and the total number of larvae + unhatched eggs counted in each test were added to the model as covariates. Additional ANCOVA models were run comparing two levels of immersion duration to complete the analysis. In addition, linear regression analyses were run for each immersion duration level (α = 0.05). LC_50_ and LC_90_ values (Lethal Concentration: the concentration of a substance that kills 50% and 90% of a test population, respectively) were calculated with a 95% confidence interval for each immersion duration level using inverse prediction for dose (ppm) at Mortality = 0.5, rounded to the nearest digit. In parallel to the ANCOVA model, a general ANOVA model was also run using chlorine concentration as a categorial variable. This latter model was run to do pairwise comparisons of egg mortality across chlorine concentration levels for each exposure period. To correct for multiple testing, a Bonferroni-corrected α-level was set at 0.001. Means were compared using a Tukey post-hoc test. Percentage egg survivorship data were transformed using the arcsine square root transformation to improve normality before analysis. To determine the impact of chlorination when accounting for natural mortality of *C. capitata* eggs, egg survivorship data were transformed with the Abbott formula [20], using egg survivorship values at chlorine concentration = 0 ppm to calculate natural egg mortality for each immersion duration. Data transformed with Abbott’s formula were used to illustrate the real impact of chlorine solutions, but statistical tests were performed with the untransformed data.

## 3. Results

Chlorine concentration and duration of immersion explained 80% of the variation in egg survivorship (*R*^2^ = 0.80, *F*_7,142_ = 87.8, *p* < 0.0001). The highest egg survivorship was observed in control eggs (0 ppm) immersed for 48 h (85.2 ± 1.2%, mean ± SE), while the lowest occurred at the highest concentration tested (65,000 ppm) after 48 h of immersion (1.5 ± 1.0%). Both chlorine concentration (*F*_1,143_ = 355.4, *p* < 0.0001) and immersion duration (*F*_2,143_ = 58.1, *p* < 0.0001) had a significant effect on egg survivorship, as well as their interaction (*F*_2,143_ = 36.2, *p* < 0.0001), indicating that the effect of chlorine concentration varied depending on immersion duration. The total number of eggs per test did not affect egg survivorship (*F*_1,143_ = 0.3, *p* = 0.61), nor did the chlorine product used (*F*_1,143_ = 0.4, *p* = 0.52). Separate linear regression models run for each immersion duration level indicated a significant negative correlation between chlorine concentration and egg survivorship:-For 4 h, % egg survivorship = 78.7 − 0.00039 × chlorine concentration, *R*^2^ = 0.39, *F*_1,48_ = 32.5, *p* < 0.0001. LC_50_ value = 71,509 ppm; LC_90_ value = 192,769 ppm.-For 24 h, % egg survivorship = 79.0 − 0.00098 × chlorine concentration, *R*^2^ = 0.88, *F*_1,48_ = 347.8, *p* < 0.0001. LC_50_ value = 27,048 ppm. LC_90_ value = 73,420 ppm.-For 48 h, % egg survivorship = 72.0 − 0.00115 × chlorine concentration, *R*^2^ = 0.76, *F*_1,48_ = 158.2, *p* < 0.0001. LC_50_ value = 16,987 ppm. LC_90_ value = 53,214 ppm.

To further investigate the interaction effect between chlorine concentration and immersion duration, three additional ANCOVA models were run, each comparing two immersion durations. In each model, the interaction between immersion duration and chlorine concentration remained significant (4 h vs. 24 h: *F*_1,96_ = 44.9, *p* < 0.0001; 4 h vs. 48 h: *F*_1,96_ = 64.4, *p* < 0.0001; 24 h vs. 48 h: *F*_1,96_ = 9.8, *p* < 0.01). These results indicate that the rate of decrease in egg survivorship with increasing chlorine concentration differed significantly for each pair of immersion duration levels. The decrease in egg survivorship was steepest when eggs were immersed for 48 h, weakest when eggs were immersed for 4 h, and intermediate when eggs were immersed for 24 h (Figure 1). The full dataset is provided in the Appendix A.

The egg mortality data adjusted for natural egg mortality with Abbott’s formula are given in Table 1 and illustrate the real impact of each dose of chlorine and immersion duration tested on *C. capitata* egg mortality (Table 1).

## 4. Discussion

The need to eliminate live insects from wastewater is a challenge faced by arthropod containment facilities worldwide. Although chlorination is occasionally used for this purpose, its efficacy as a universal insect eradication method has not yet been thoroughly investigated. Our study demonstrates that chlorine is ineffective at eradicating the eggs of the dipteran pest *Ceratitis capitata*; even extremely high concentrations of sodium hypochlorite and calcium hypochlorite (up to 65,000 ppm) failed to achieve 100% egg mortality. The two chlorine products tested (sodium hypochlorite and calcium hypochlorite) produced statistically similar effects, showing the consistency of this result.

There was a clear negative linear relationship between chlorine concentration and *C. capitata* egg survivorship, and longer immersion durations accentuated the effects of chlorine. These patterns are consistent with previous studies investigating the effects of chlorine on mosquitoes and chironomid larvae [7,8]. However, the concentrations at which chlorine was effective in those studies drastically differ from the values we observed. Hidayaturrahman et al. (2023) [8] reported > 80% mortality of chironomid larvae when immersed in water containing 20 ppm of chlorine for 5 days. Shahen et al. (2020) [7] reported doses of 61 ppm of calcium hypochlorite and 301 ppm of sodium hypochlorite were needed to kill 90% of 4th-instar mosquitos with a 24 h exposure period. In contrast, our study required approximately 30,000 ppm of chlorine to reach 50% mortality of *C. capitata* eggs with an immersion duration of 24 h (Figure 1). We conclude that eggs of *C. capitata* are extremely resistant to chlorine compared to the larvae of the aquatic insects that have been the focus of previous studies. The doses needed to kill most *C. capitata* eggs would be completely impractical to apply for wastewater treatment in arthropod containment buildings and would lead to the formation and release of high levels of chlorine byproducts harmful to the environment [4,21]. The results obtained with *C. capitata* are reminiscent of other organisms that are infamously resistant to chlorine and other disinfectants, such as eggs of parasitic roundworms (e.g., *Ascaris suum*) and protozoan oocysts (e.g., *Cryptosporidium* spp.) [22,23]. Rather than being used to eliminate eggs of *C. capitata*, chlorine could potentially be used to disinfect *C. capitata* eggs without harming them for rearing or research purposes [17].

The mechanisms behind the high chlorine resistance of *C. capitata* eggs deserve further investigation. In contrast to water-dwelling insects such as mosquito larvae, which actively ingest or absorb water through feeding and osmosis, *C. capitata* eggs remain largely inert when immersed, as water significantly slows their development. This suggests that ingestion or absorption might be required for chlorine to exert toxicity at low concentrations, and that external application alone may be insufficient. A few studies have specifically documented the efficacy of house bleach as an ovicide against mosquitoes, seemingly contradicting this argument [5,24,25]. However these studies either measured ovicidal success indirectly by assessing the presence or absence of larvae in treated water, leaving it unclear whether the eggs or the hatching larvae were affected [24], or used very high doses of chlorine (e.g., 20–40 ppt) [5]. It should also be noted that chlorine products are known to remove the external layer of the chorion of various insect eggs, a process known as dechorionation [24,26]. There is therefore an undisputable effect of chlorine on the chorion. However, the doses needed to compromise the embryo itself and cause egg mortality are likely much higher and may vary substantially among species. Future research exploring the possible biological mechanisms explaining the remarkable tolerance of *C. capitata* eggs should focus on assessing the impermeability of the chorion, the lack of metabolic activity during immersion, and the protective proteins or lipid layers present on the inner layers of the chorion that may reduce chemical penetration, and compare these measures to those of other organisms known for their resistance or lack of resistance to chlorine.

Although the effects of chlorine exposure during the egg stage on larval fitness were not the focus of this study, they should not be underestimated. Negative post-hatching effects of exposure to sublethal doses of insecticides during the egg stage have been documented in other systems [27]. Such effects could decrease the likelihood of survival and establishment of insects originating from eggs present in chlorinated water. As these hypothetical effects could be species-specific, they should be investigated on a case-by-case basis.

The linearity of the negative correlation between chlorine concentration and egg survivorship that was observed for each exposure duration is somewhat surprising. Indeed it has been shown that very high concentrations of chlorine (e.g., >20,000 ppm) have a paradoxically lower oxidative power than lower concentrations [28]. At very high concentrations, chlorine undergoes reactions that convert it into less reactive oxidants and reduce the amount of free available chlorine. Therefore, it could have been expected to see the effects of chlorination on *C. capitata* egg mortality reach a plateau before decreasing at very high concentrations. Instead, the highest egg mortality was always observed at the highest chlorine concentration for each exposure duration. However, because solution pH tends to increase with chlorine concentration [28], an alternative explanation is that part of the observed egg mortality may result from the effects of high alkalinity rather than chlorine itself [29]. These considerations highlight the need for future studies to quantify chemical parameters such as pH and oxidation-reduction potential (ORP) to disentangle the relative contributions of chlorine concentration and solution alkalinity to insect egg mortality. These parameters are important for understanding chlorine stability and speciation, particularly at high concentrations.

Although the regulatory frameworks that oversee arthropod containment facilities generally mandate the elimination of live organisms from wastewater [10,11], they may lack clear guidance on how to achieve this goal. In practice, many facilities continue to rely on chlorine disinfection due to its convenience and widespread availability. This continued reliance on chlorine disinfection is highly concerning because this treatment does not effectively eliminate the risk of dissemination of all live arthropods in the environment and the repeated use of high doses of chlorine may have negative consequences for the environment and human health. There is an urgent need for a thorough evaluation of chlorine’s effectiveness across different arthropod groups and developmental stages to determine the conditions under which this approach could be considered appropriate for insect elimination. In the meantime, more reliable and sustainable alternatives to eliminate insects from water effluents such as heat treatment [30], mechanical water filtration [11], or integrated multi-barrier approaches should be prioritized for wastewater management in containment facilities to ensure both biosafety and environmental protection.

## Figures and Tables

**Figure 1 insects-16-01213-f001:**
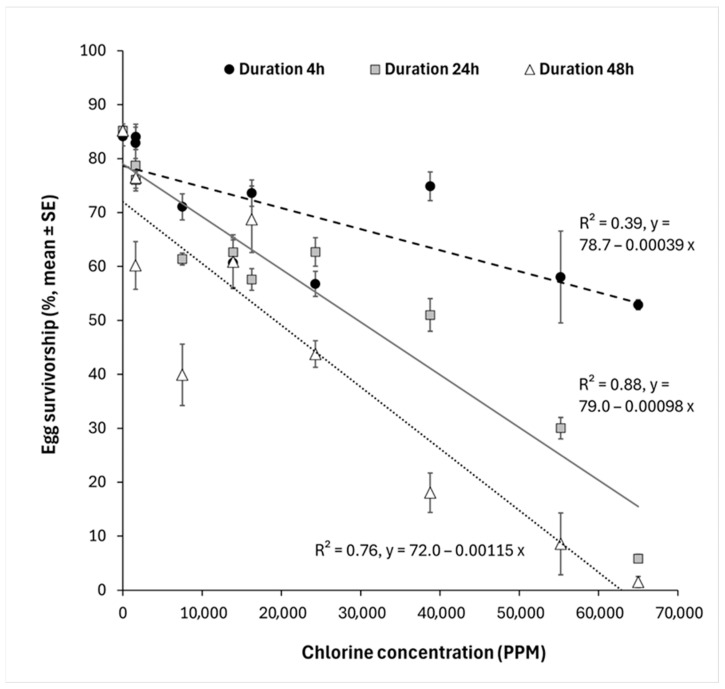
Effects of different concentrations of chlorine expressed in parts per million (PPM) on the survivorship of eggs of Ceratitis capitata (%, mean ± SE, *n* = 5 for each dot) for three different durations of immersion in a chlorine solution (4, 24, and 48 h). Fitted lines show linear relationships between chlorine concentration and egg survivorship for each immersion duration (4 h: dashed line; 24 h: plain line; 48 h: dotted line). R2 values and equations are given for each regression line.

**Table 1 insects-16-01213-t001:** Impact of different chlorine concentrations (PPM) on *Ceratitis capitata* egg mortality, after adjusting for natural mortality using Abbott’s formula (mean ± SE, *n* = 5 for each cell). Standard error values were calculated based on the untransformed standard error values with the delta method, using a first-order Taylor approximation. Within each column, means sharing different letters are significantly different (Tukey post-hoc test, Bonferroni-corrected α = 0.001). Pairwise comparisons should be interpreted with caution due to the low number of replicates per group (*n* = 5).

Chlorine Concentration (PPM)	Immersion Duration 4 h	Immersion Duration 24 h	Immersion Duration 48 h
0	0 ± 3.0 a	0 ± 1.3 a	0 ± 2.2 a
1600–1625	0.7 ± 3.8 a	9.2 ± 3 a	19.7 ± 4.2 ab
7490	15.6 ± 3.4 ab	27.9 ± 1.5 b	53.2 ± 6.7 bc
13,900	27.8 ± 5.7 b	26.5 ± 2.8 b	28.5 ± 6.0 ab
16,250	12.5 ± 3.4 ab	32.4 ± 2.4 b	19.4 ± 7.4 ab
24,280	32.5 ± 3.1 b	26.4 ± 3.2 b	48.6 ± 2.9 bc
38,760	10.9 ± 3.7 ab	40.1 ± 3.6 b	78.9 ± 4.2 cd
55,220	31.0 ± 10.2 b	64.8 ± 2.4 c	89.9 ± 6.7 d
65,000	37.1 ± 1.7 b	93.2 ± 0.9 d	98.2 ± 1.2 d

## Data Availability

The original contributions presented in this study are included in the article/Appendix A. Further inquiries can be directed to the corresponding author.

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
