# Peer review of "Chlorination Is Ineffective at Eliminating Insects from Wastewater: A Case Study Using Ceratitis capitata"

_insects, 2025, doi:10.3390/insects16121213_

Round 1

Reviewer 1 Report

Comments and Suggestions for Authors

This study presents a well-defined research question, focusing on the effects of chlorination in water treatment and its potential impact on aquatic insects. Beyond its primary role in water purification, the research also provides insights into the possible use of chlorination for insect control or for assessing its ecological effects on insects and other aquatic organisms.

However, there are some weaknesses that need to be improved as follows:

  1. Please clarify the number of eggs used in each treatment and replication. The total sample size is stated as n = 150 — does this mean 30 eggs per replication and 150 eggs for five replications in each treatment?
  2. Please provide statistical comparisons (or indicate significant differences) in Table 1.
  3. It will be usefull if you have the results of the test with standard chlorination treatment in this study in order to show that for the standard used of chlorination treatment is not effect this insect egg.
  4. It would be useful if you included results from a standard chlorination treatment in this study to demonstrate that the typical chlorination level has no effect on the insect eggs.
  5. Although this study has a good and interesting research question, the content is rather limited. I recommend adding more experimental work and expanding the discussion of the results

5.1 the detail of LC50 may be provide

5.2 the survival of larval stage of the survival egg.

5.3 Explain the response mechanism of this insect’s eggs, including their tolerance to environmental conditions and whether they undergo dormancy. Compare these characteristics with mosquitoes and freshwater midges studied previously.

5.4 It would be more interesting if you could include a figure showing abnormal eggs compared with normal ones.

Author Response

[comment 1]This study presents a well-defined research question, focusing on the effects of chlorination in water treatment and its potential impact on aquatic insects. Beyond its primary role in water purification, the research also provides insights into the possible use of chlorination for insect control or for assessing its ecological effects on insects and other aquatic organisms.

[response 1] Thanks for your comments and feedback on our study.

[comment 2]However, there are some weaknesses that need to be improved as follows:

Please clarify the number of eggs used in each treatment and replication. The total sample size is stated as n = 150 — does this mean 30 eggs per replication and 150 eggs for five replications in each treatment?

[response 2] Thanks for this comment, please let us clarify. N = 150 is the total number of tests conducted. There were 5 replications for each dose (10 doses in total) and exposition duration (3 durations in total) tested, so the total number of tests was 5 * 10 * 3 = 150. In each test we used a fixed quantity of 1 mL of a water solution containing C. capitata eggs. Eggs of C. capitata are very fragile and it would have been very difficult to manipulate individual eggs. Instead, eggs were directly collected from the rearing in water and 1mL was taken from that solution and used for each test. The number of eggs contained in 1 mL of solution was 88.3 ± 3.3. In total, we used 13 249 eggs across the 150 tests. These information have  been clarified in the material and methods section of the revised manuscript.

[comment 3] Please provide statistical comparisons (or indicate significant differences) in Table 1.

[Response 3] We appreciate the reviewer’s suggestion to include statistical comparisons in table 1, which implies conducting pairwise comparisons among all doses tested. In our design, doses represent a continuous predictor within a dose–response framework, and the experiment was planned for ANCOVA modeling rather than categorical treatment comparisons. With 5 replicates per dose, we considered that pairwise tests across 10 doses (45 comparisons) would have limited statistical power after correction for multiple testing, and opted to present the modeled dose–response relationship, which more appropriately reflect the design and aims of the study.

Nonetheless, to address the reviewer’s concern, we have included a new version of the table with pairwise comparisons (after applying a Bonferroni correction to the alpha level of the model) in the manuscript, and added to the legend of the table the following statement “Pairwise comparisons should be interpreted with caution due to the low number of replicates per group (n = 5).”

[comment 4]It will be usefull if you have the results of the test with standard chlorination treatment in this study in order to show that for the standard used of chlorination treatment is not effect this insect egg.

[response 4] We do not have quantified data of these early trials with standard chlorination treatments (presence/absence of live larvae was the variable measured). However, we now cite a paper in this section that state that standard chlorination can be used to clean C. capitata eggs but does not harm them (Zaada, D.S.Y.; Ben-Yosef; M., Yuval, B.; Jurkevitch, E., 2019. The host fruit amplifies mutualistic interaction between Ceratitis capitata larvae and associated bacteria). 

[comment 5]It would be useful if you included results from a standard chlorination treatment in this study to demonstrate that the typical chlorination level has no effect on the insect eggs.

[response 5] Please see response above.

[comment 6] Although this study has a good and interesting research question, the content is rather limited. I recommend adding more experimental work and expanding the discussion of the results

[response 6] Thank you for your comment. While we recognize that the dataset is limited, the main result—that chlorination is ineffective at eliminating Ceratitis capitata from wastewater—is robust and has important practical implications for arthropod containment facilities. Given the clarity and significance of this finding, we intentionally focused on presenting a concise study suitable for rapid communication. Expanding the experimental scope would be valuable for future work, but we believe the current data sufficiently support the key conclusion. We should also add that the rearing of C. capitata has been stopped a year ago in our laboratory and that additional experimental work on this research topic is unfortunately not possible at the moment.

Nonetheless, to address the reviewer’s concern, we expanded the discussion of the results as suggested in the revised version of the manuscript.

[comment 7]5.1 the detail of LC50 may be provide

[response 7] We have now added the LC50 and LC90 estimates with 95% CI in the results section as well as how they were estimated in the M&M section.

[comment 8] 5.2 the survival of larval stage of the survival egg.

[response 8] This is indeed an interesting point. We have now mentioned the potential post-hatching effects of chlorine treatments in the discussion.

[comment 9]5.3 Explain the response mechanism of this insect’s eggs, including their tolerance to environmental conditions and whether they undergo dormancy. Compare these characteristics with mosquitoes and freshwater midges studied previously.

[response 9] We agree that such data would be highly valuable and could be the subject of further studies. This is reinforced in the discussion section of the revised manuscript.

[comment 10]5.4 It would be more interesting if you could include a figure showing abnormal eggs compared with normal ones.

[response 10] Unfortunately we did not take pictures of the unhatched (“abnormal”) eggs and since the rearing has ended we will not be able to produce pictures that could serve as illustration in this manuscript.

Reviewer 2 Report

Comments and Suggestions for Authors

Dear Authors,

The manuscript has been found to be of high scientific quality, demonstrating both originality and robustness. It addresses a relevant and underexplored issue in biosafety and wastewater management — namely, the persistence of insect eggs under standard chlorination conditions.
Following minor revisions concerning methodological clarification, data presentation, and contextual depth, the manuscript will be ready for publication.
My specific comments are as follows:

1. Chemical Characterisation of Chlorine Solutions

The study would be strengthened by including empirical data on pH, residual chlorine concentration, and oxidation–reduction potential (ORP). These parameters are essential for interpreting chlorine stability and reactivity, particularly at high concentrations where chlorine speciation (HOCl/OCl⁻ balance) may vary significantly.

If such measurements were not performed, please provide a brief discussion of the implications of their absence, emphasizing how they might influence chlorine persistence and the observed biological effects.

2. Mechanism of Egg Resistance

The discussion should explore possible biological mechanisms that could explain the remarkable tolerance of Ceratitis capitata eggs.
Consider mentioning:

  • The impermeability of the chorion,

  • The lack of metabolic activity during immersion,

  • And potential protective proteins or lipid layers that reduce chemical penetration.

This would enhance the biological interpretation of your findings and connect them more broadly to insect physiology.

3. Comparative Sensitivity Across Taxa and Life Stages

It would benefit the discussion to include a brief comparative note on chlorine sensitivity among other insect species or life stages (e.g., larvae, pupae).
This comparison would help readers generalize the implications of your findings for containment protocols and wastewater treatment practices in different taxa or rearing systems.

4. Figures and Regression Modelling

  • Please ensure that Figure 1 includes error bars (standard deviation or standard error) and that regression equations with R² values are presented directly on the plot.

  • You may wish to explore non-linear models (e.g., log-logistic or exponential dose–response curves) to capture possible biological plateaus at low chlorine concentrations.

  • Include standard deviations or confidence intervals in both Figure 1 and Table 1 for statistical transparency.

5. Simple Summary

Consider revising the Simple Summary to include a succinct statement on the environmental risks associated with chlorine by-products and the rationale for exploring non-chemical or alternative disinfection methods such as UV, ozone, or thermal treatment.

6. Chlorine Source Consistency

Please clarify whether both chlorine sources (NaOCl and Ca(OCl)₂) produced statistically similar results.
If differences were detected, briefly describe them and discuss their potential implications for real-world applications.

7. Supplementary Material and Data Transparency

  • The meaning of the ‘Treatment’ column (currently labelled “Co”) in the supplementary Excel file should be clearly defined.

  • For full transparency and reproducibility, consider providing the raw dataset and/or analysis code (e.g., JMP, SAS, or R script) as an additional supplementary file.

8. Minor Language and Formatting Adjustments

Please review the text for typographical consistency. For instance:

  • Replace “chlorination-associated” (L46) with the corrected hyphenation.

  • Replace “de-chorionation” (L218) with “dechorionation.”
    Ensure consistency in number formatting (thousand separators, decimals) and define all abbreviations (e.g., SIT, EBCL) at first mention.

9. References

The current reference list is strong and relevant. It could be further enhanced by including recent reviews on non-chlorine disinfection technologies, such as ultraviolet (UV) light, ozone, and heat treatment, to strengthen the environmental and applied dimension of the discussion.

Author Response

Dear Authors,

The manuscript has been found to be of high scientific quality, demonstrating both originality and robustness. It addresses a relevant and underexplored issue in biosafety and wastewater management — namely, the persistence of insect eggs under standard chlorination conditions. Following minor revisions concerning methodological clarification, data presentation, and contextual depth, the manuscript will be ready for publication.

My specific comments are as follows:

[comment 1] 1. Chemical Characterisation of Chlorine Solutions

The study would be strengthened by including empirical data on pH, residual chlorine concentration, and oxidation–reduction potential (ORP). These parameters are essential for interpreting chlorine stability and reactivity, particularly at high concentrations where chlorine speciation (HOCl/OCl⁻ balance) may vary significantly.

If such measurements were not performed, please provide a brief discussion of the implications of their absence, emphasizing how they might influence chlorine persistence and the observed biological effects.

[response 1] We appreciate the reviewer’s comment and this new perspective. We did not measure pH, residual chlorine concentration, and oxidation–reduction potential (ORP) in our study because the focus of the study was on the direct impact of chlorination on insect mortality rather than on the stability of chlorine when used at high concentrations. However, it is correct that chlorine is likely to become paradoxically less effective at very high concentrations because less active chlorine is available. Our dataset shows a very linear relationship between chlorine concentration and egg mortality, showing that chlorine efficiency at the very least did not decrease at high concentrations. It should also be considered that ph increases with chlorine concentration, and that highly alkaline solutions have been shown to impact insect eggs. Therefore, this opens the door to a new interpretation: the mortality of C. capitata eggs could be due to increasing ph independently of the presence of chlorine. We have included these new considerations in the discussion of the manuscript.

[comment 2]

  1. Mechanism of Egg Resistance

The discussion should explore possible biological mechanisms that could explain the remarkable tolerance of Ceratitis capitata eggs.

Consider mentioning:

The impermeability of the chorion, the lack of metabolic activity during immersion, and potential protective proteins or lipid layers that reduce chemical penetration. This would enhance the biological interpretation of your findings and connect them more broadly to insect physiology.

[response 2] Thanks for this comment. We have expanded the mentions of these mechanisms in the discussion of the revised manuscript.

[comment 3]

  1. Comparative Sensitivity Across Taxa and Life Stages

It would benefit the discussion to include a brief comparative note on chlorine sensitivity among other insect species or life stages (e.g., larvae, pupae). This comparison would help readers generalize the implications of your findings for containment protocols and wastewater treatment practices in different taxa or rearing systems.

[response 3] We do agree with the reviewer. The impact of chlorine on insects in water is unfortunately poorly documented and we have already cited most of the existing literature on the topic in the manuscript. A broader exploration of chlorine effects on different insects species and life stages simply does not exist in the scientific literature at the moment.

[comment 4]

  1. Figures and Regression Modelling

Please ensure that Figure 1 includes error bars (standard deviation or standard error) and that regression equations with R² values are presented directly on the plot.

[response 4] Error bars, regression equations, and R² values were included in figure 1 in the original version of the manuscript and standard errors measures were included in the table on the original manuscript.

[comment 5] You may wish to explore non-linear models (e.g., log-logistic or exponential dose–response curves) to capture possible biological plateaus at low chlorine concentrations.

[response 5] We did explore non-linear models but, somewhat surprisingly, linear models gave the best results in terms of model fitting.

[comment 6] Include standard deviations or confidence intervals in both Figure 1 and Table 1 for statistical transparency.

[response 6] Error bars, regression equations, and R² values were included in figure 1 in the original version of the manuscript and standard errors measures were included in the table on the original manuscript.

[comment 7] Simple Summary

Consider revising the Simple Summary to include a succinct statement on the environmental risks associated with chlorine by-products and the rationale for exploring non-chemical or alternative disinfection methods such as UV, ozone, or thermal treatment.

[response 7] Thanks for this comment, we have now expanded our mentions of the environmental risks associated with chlorine by products and alternative disinfection methods to eliminate insects from water in the introduction and discussion sections of the manuscript.

[comment 8] 6. Chlorine Source Consistency

Please clarify whether both chlorine sources (NaOCl and Ca(OCl)₂) produced statistically similar results. If differences were detected, briefly describe them and discuss their potential implications for real-world applications.

[response 8] As stated in the results section, the two chlorine products produced similar results. We have expanded on this statement in the discussion section of the revised manuscript.

[comment 9]7. Supplementary Material and Data Transparency

The meaning of the ‘Treatment’ column (currently labelled “Co”) in the supplementary Excel file should be clearly defined.

[response 9] Thanks for this comment. We have now changed “Co” and replace it with “Control”, which should be much more clear.

[comment 10] For full transparency and reproducibility, consider providing the raw dataset and/or analysis code (e.g., JMP, SAS, or R script) as an additional supplementary file.

[response 10] The software used for the analyses in mentioned in the M&M section (JMP 18). The script of the different analyses is now provided in the supplementary files.

[comment 11]. Minor Language and Formatting Adjustments

Please review the text for typographical consistency. For instance: Replace “chlorination-associated” (L46) with the corrected hyphenation.

[response 11] This has been replaced.

[comment 12] Replace “de-chorionation” (L218) with “dechorionation.”

[response 12] This has been replaced.

[comment 13] Ensure consistency in number formatting (thousand separators, decimals) and define all abbreviations (e.g., SIT, EBCL) at first mention.

[response 13] We have made sure that all abbreviations were defined at their first mention.

[comment 14] 9. References

The current reference list is strong and relevant. It could be further enhanced by including recent reviews on non-chlorine disinfection technologies, such as ultraviolet (UV) light, ozone, and heat treatment, to strengthen the environmental and applied dimension of the discussion.

[response 14] We have added a few new references in the revised manuscript, following the suggestions of all three reviewers.

Reviewer 3 Report

Comments and Suggestions for Authors

Dear Editor:

  • I reviewed the manuscript entitled “Chlorination is ineffective at eliminating insects from wastewater: a case study using Ceratitis capitata”, this explain the effect of chlorination in the survivor of C. capitata eggs with different concentrations and time of exposition. The information is important and the results and are adequately discussed. Inclusive, the introduction and the material and methods are good presented.
  • I have only some important questions:
  • I did not understand Why did you use C. capitata eggs, this insect it is an important pest, but need to develop into some fruit. So, in the wastewater I think impossible to maintain eggs of C. capitata, or they do not represent any risk as pest.
  • Other question is, whether the authors Do they want to use C. capitata eggs as a model for analyze the effects of chlorination. If this is the case, they should include other comments in the discussion.
  • Curiously, the information can be useful for to clean eggs in mass rearing of C. capitata. However, the authors need to discuss also this subject.
  • In other words, this work has interesting evaluations, but has not an important justification.

Author Response

[comment 1]Dear Editor:

I reviewed the manuscript entitled “Chlorination is ineffective at eliminating insects from wastewater: a case study using Ceratitis capitata”, this explain the effect of chlorination in the survivor of C. capitata eggs with different concentrations and time of exposition. The information is important and the results and are adequately discussed. Inclusive, the introduction and the material and methods are good presented.

[response 1] Thanks for your feedback on our manuscript.

[comment 2]I have only some important questions:

I did not understand Why did you use C. capitata eggs, this insect it is an important pest, but need to develop into some fruit. So, in the wastewater I think impossible to maintain eggs of C. capitata, or they do not represent any risk as pest.

[response 2] Thanks for this comment. We agree with the reviewer that C. capitata eggs are highly unlikely to develop into adults if they hatch in water, far from their host fruits. However, 1) the focus of the study is to show that chlorination should not be considered as a universal method guaranteeing successful eradication of insects in wastewater. Some insect quarantines unfortunately still rely to this day on this method to say that their wastewater is “insect-free”. Showing that a single insect species is resistant to chlorination is enough to prove the point that is method is far from an universal effective insecticide, regardless of the true potential for escape of the species and life stage targeted. And 2) Many tephritid species, not only C. capitata, are reared in arthropod rearing facilities because they are important pests worldwide. It is therefore an important group to consider when investigating insects that could be found in the wastewater of these facilities. The real potential for escape from wastewater is likely to be species-dependent and should be investigated on a case-by-case basis.

[comment 2]Other question is, whether the authors Do they want to use C. capitata eggs as a model for analyze the effects of chlorination. If this is the case, they should include other comments in the discussion.

[response 2] Please see above response, which we think addresses this comment. We have revised the introduction section of the manuscript to justify the use of C. capitata as a model species.

[comment 3] Curiously, the information can be useful for to clean eggs in mass rearing of C. capitata. However, the authors need to discuss also this subject.

[response 3] Indeed, chlorination has been proposed as a way to clean C. capitata eggs for research purposes. We have added the reference of this work in the introduction of the revised version of the manuscript. “Zaada, D.S.Y.; Ben-Yosef; M., Yuval, B.; Jurkevitch, E., 2019. The host fruit amplifies mutualistic interaction between Ceratitis capitata larvae and associated bacteria”. We also briefly discuss this subject in the revised discussion section of the manuscript.

[comment 4]In other words, this work has interesting evaluations, but has not an important justification.

[response 4] Thanks for your feedback on our manuscript. We hope that the revised version of the manuscript strengthens the justification for the study and improves the overall cohesion of the manuscript.

Round 2

Reviewer 1 Report

Comments and Suggestions for Authors

The author has addressed most of the manuscript’s details, which has strengthened the paper.

Reviewer 3 Report

Comments and Suggestions for Authors

Dear Editor:

The last presented form in my opinion complete all recommendations. I do not have any questions.

The manuscript must be accepted.

Congratulations.